# Acacetin Inhibits Cell Proliferation and Induces Apoptosis in Human Hepatocellular Carcinoma Cell Lines

**DOI:** 10.3390/molecules27175361

**Published:** 2022-08-23

**Authors:** Manal Alfwuaires, Hany Elsawy, Azza Sedky

**Affiliations:** 1Department of Biological Sciences, College of Science, King Faisal University, Al Ahsa 31982, Saudi Arabia; 2Department of Chemistry, College of Science, King Faisal University, Al Ahsa 31982, Saudi Arabia; 3Department of Chemistry, Faculty of Science, Tanta University, Tanta 31527, Egypt; 4Department of Zoology Faculty of Science, Alexandria University, Alexandria 21526, Egypt

**Keywords:** hepatocellular carcinoma, acacetin, STAT3, proliferation, apoptosis

## Abstract

Human hepatocellular carcinoma (HCC) is the fifth most common cancer and the third leading cause of death across the world. Recent evidence suggests that STAT3 regulates proliferative, survival, metastasis, and angiogenesis genes in HCC. Novel agents that suppress STAT3 activation can be used to prevent or treat HCC. We used a functional proteomics tumor pathway technology platform and multiple HCC cell lines to investigate the effects of acacetin (ACN) on STAT3 activation, protein kinases, phosphatases, products of STAT3-regulated genes, and apoptosis. ACN was found to inhibit STAT3 activation in a dose- and time-dependent manner in HCC cells. Upstream kinases c-Src, Janus-activated kinase 1, and Janus-activated kinase 2 were also inhibited. The ACN inhibition of STAT3 was abolished by vanadate treatment, suggesting the involvement of tyrosine phosphatase activity. ACN was found to suppress the protein expression of genes involved in proliferation, survival, and angiogenesis via STAT3 inhibition. ACN appears to be a novel STAT3 inhibitor and may be a promising therapeutic compound for application in the treatment of HCC and other cancers.

## 1. Introduction

STAT3 proteins play an important role in the survival and proliferation of tumor cells [1]. As a latent factor of transcription, STAT3 is situated within the cytoplasm [2,3]. Following activation by cytokines, such as interleukin, or growth factors derived from EGF platelets, STAT3 undergoes phosphorylation-induced homodimerization, resulting in gene transcription, which involves DNA binding, modulation, and translocation. Mediation of STAT3 phosphorylation takes place after the activation of Janus-like kinase, a protein (non-receptor) in the tyrosine kinase family. This family includes TYK2/JAK1/JAK2/JAK3 [4] responsible for STAT3 (Tyr705) phosphorylation [5]. In normal cells, the duration for which STAT3 is activated remains transient, making a significant contribution to cell proliferation and organ development. However, the involvement of constitutive STAT3 activation is common in tumors [6,7], including in the case of hepatocellular carcinoma (HCC). STAT3, in turn, reportedly enhances cellular proliferation by modulating the expression of various genes necessary for tumor cells to survive, grow, and undergo metastasis/angiogenesis [8,9]. STAT3 increases the levels of anti-apoptotic proteins (e.g., Bcl-xL, Bcl-2, survivin, and Mcl-1) and generates pro-proliferative signals [10,11,12]. Moreover, STAT3 can regulate the expression of VEGF, a growth factor necessary for angiogenesis and tumor vasculature maintenance [13,14]. Finally, STAT3 has been shown to block the expression of inflammatory factors during tumor growth [15]. Thus, targeting STAT3 phosphorylation using a small-molecule approach could be a potential strategy for chemoprevention and cancer therapy.

To identify natural-source-derived pharmacological drugs that could be useful for cancer therapy by targeting STAT3 activation pathways, focus has been placed on acacetin isolated from plants, such as Tunera diffusa, Propolis, Dracocephalum moldavica, Betula pendula, Robinia pseudoacacia, Flos Chrysanthemi Indici, Chrysanthemum, Safflower, Calamintha, and Linaria species [16,17]. As a natural flavonoid compound, acacetin (5,7-dihydroxy-4′-methoxyflavone) has antioxidative and anti-inflammatory effects and has exhibited extensive therapeutic effects on numerous cancers [17,18,19,20,21]. However, the elaborate molecular mechanisms through which ACN brings about these effects and its potential in vitro efficacy on HCC progression remain unclear. The effect of ACN on activation of STAT3 and associated protein kinases and phosphatase, STAT3-regulated gene products, cellular proliferation, and apoptosis was investigated in cells with constitutive (HepG2) and inducible (Huh-7) STAT3 activation.

## 2. Materials and Methods

### 2.1. Reagents and Cell Culture

Acacetin (purity > 98%), bovine serum albumin, SDS, MTT, NaCl, glycine, and goat anti-rabbit horseradish peroxidase (HRP) conjugate were purchased from Sigma-Aldrich. An ACN stock solution (1 mol/mL) was prepared in DMSO and stored at −20 °C for use within 3 months of preparation. This ACN stock solution was further diluted in DMEM, also referred to as Dulbecco’s modified Eagle’s medium, to prepare a working solution that was further finally diluted in the experimental cell culture medium. Blue vital stain, 0.4% trypan, FBS, and antibiotic–antimycotic mixture, as well as DMEM, were acquired from Invitrogen. Mouse monoclonal as well as rabbit polyclonal antibodies were obtained from Santa Cruz Biotechnology and used against the following targets: Akt, Bcl-xL, survivin, D1, VEGF, caspase3, Mcl-1, cyclin, PARP, phospho-STAT3, and phospho-Akt. Antibodies for Src, Tyr 416, JAK2, Tyr 1007/1008, JAK1, and Tyr 1022/1023 were acquired from Cell Signaling Technology. Ki67 was obtained from BD Biosciences PharMingen. The DNA-binding/nuclear extraction kits were acquired from Active Motif and the scrambled control siRNAs from Santa Cruz Biotechnology. Recombinant human IL-6 in bacteria was obtained from ProSpec-Tany TechnoGene Ltd. (Hamada, Rehovot, Israel).

### 2.2. Cell Lines

HepG2 as well as Huh-7 human HCC cell lines were obtained from American Type Culture Collection. These cell lines cultured in DMEM containing 1× antibiotic–antimycotic and 10% FBS.

### 2.3. MTT Assay

HepG2 and C3a cells (1 × 10^4^ cells/mL) were cultured in 96-well plates and incubated with various concentrations of ACN (purity 98%, Sigma-Aldrich (St. Louis, MO, USA) for 24, 48, and 72 h, respectively, and 3-(4,5-dimethylthiazol-2-yl)-2,5-diphenyltetrazolium bromide, also known as 2,5-diphenyl-2H-tetrazolium bromide (MTT)), was applied to determine cell viability on the basis of the manufacturer’s protocol. The viability was calculated by comparing the absorbance of each group with control cells, which were arbitrarily assigned a value of 100%.

### 2.4. Cell Invasion

We used BD Matrigel invasion chambers to test invasion. In the invasion assay, 10 µL of Matrigel was applied to a polycarbonate membrane filter, and the standard medium was added at the bottom of the apparatus. Cells were seeded in top chambers, with 1 × 10^5^ cells in 500 µL of a serum-free medium, and ACN was added in the indicated doses. In the bottom chambers, we added 750 µL of 10% FBS to the media. Cells were incubated for 24 h to allow migration. After incubation with a cotton swab, we removed the non-migrated cells on top of the membrane. Migrated cells were fixed in cold 75% methanol for 15 min and then washed three times in PBS. After Giemsa staining, the cells were again washed with PBS. Images were taken with an optical microscope (200× magnification), and invading cells were manually counted. Invading cells were quantified as a percentage of inhibition, with control cells assigned a value of 100%.

### 2.5. Western Blotting

Protein was extracted from gastric cancer cells and tumor tissue samples by incubation in radioimmunoprecipitation assay (RIPA) buffer for 50 min at 4 °C. Next, we centrifuged the lysates for 15 min at 17,500× *g* at 4 °C, as previously described [22]. After centrifugation, the supernatant was used for Western blotting analysis. Protein concentration was quantified using a bicinchoninic acid (BCA) analysis kit (Beyotime, Shanghai, China). Protein lysates containing equal amounts of protein (50 μg) were separated on 10–12.5% SDS-polyacrylamide gels and transferred to nitrocellulose membranes. The membranes were subsequently blocked, incubated with specific primary antibodies (caspase-3, PARP, p-STAT3, STAT3, Bcl2, pJAK2, JAK2, psrc, src, Cuclin-D1, Bcl-xl, Survivin, VEGF, β-actin, pNF-κB, pIKBα, NF-κB, IKBα, and Cleaved caspase-3), and HRP-conjugated secondary antibodies. β-Actin was an internal loading control. The samples were examined using a LI-COR chemiluminescence imaging system (3600-00-C-Digit Blot Scanner) to visualize the protein bands. To generate the graphs, we used LI-COR Biosciences Image Studio Lite software (Lincoln, NE, USA) with a normalization of intensity according to the untreated control band, which was set to 1. Each experiment was repeated at least three times.

### 2.6. Immunofluorescence for Nuclear Translocation

After treating the cells with ACN (15 µmol) for 2 h, we fixed the cells in 4% paraformaldehyde (PFA) in 1× PBS for 20 min. After being permeabilized with 0.2% Triton X-100 for 10 min, the HepG2 cells were blocked with 5% BSA in 1× PBS for 1 h and then incubated with STAT3 primary antibody for 24 h. The next day, the cells were incubated with secondary antibodies Alexa Fluor 594 rabbit anti-rabbit IgG (H + L) for 60 min at room temperature. The nuclei of the control cells were probed with 1 μg/mL DAPI for 3 min. Measurements were performed using a Leica D6000 fluorescence microscope (Leica, Wetzlar, Germany).

### 2.7. Extraction of Cytoplasmic and Nuclear

This procedure was undertaken as per previous descriptions [23]. Cells were washed and collected in 1× hypotonic buffer. After 25 μL of detergent was added, the cytoplasmic fraction supernatant was collected following centrifugation at 14,000× *g*. Complete lysis buffer was used to resuspend the nuclear pellet. The nuclear fraction supernatant was collected and stored at −80 °C after vortexing and centrifugation.

### 2.8. DNA Binding Assay

A STAT3 ELISA kit for DNA binding was used to assay DNA binding. Nuclear extracts (5 µg) taken from cells treated with ACN was incubated inside a 96-well culture plate. Then, a specific primary antibody was used to detect STAT3. Thereafter, a secondary HRP-conjugated antibody was implemented to trace the primary antibody so as to provide a framework for the quantification by colorimetry. A microplate reader was used to measure enzymatic product formation at 450 nm. Before adding the nuclear extract, we used STAT consensus oligonucleotide (mutated/wild-type) to examine the specificity of this assay in control wells that were mutated/competitive.

### 2.9. DNA Fragmentation

Following the protocols in previous studies, cellular DNA fragments were detected using a cell death detection ELISAPLUS kit (Roche Molecular Biochemicals, Mannheim, Germany) [24].

### 2.10. Real-Time Analysis

We extracted total RNA using TRIzol and performed PCR as previously described [25].

### 2.11. TUNEL Assay

DNA fragmentation was measured using deoxynucleotidyl oligonucleotide TUNEL assays. After BSN treatments, the apoptotic cells (2 × 10^4^ in eight-well plates) were collected, fixed, and mounted in 4% formaldehyde. Hydrogen peroxide (3%) and endogenous peroxidase in methanol were used with TBS-fixed cells permeable to 20 g/mL proteinase K. Incubation with enzyme was conducted at 37 °C for 1.5 h, and the 3′OH ends of DNA were detected as a measure of apoptosis. The slides were then incubated with streptavidin and 3,3′-diaminobenzidine. The fragmented DNA of fluorescent nuclei were viewed and classified using a fluorescence microscope.

### 2.12. Statistics

Data are presented as the mean ± the SD (standard deviation) of at least three separate experiments. One-way ANOVA was used with GraphPad Prism software version 6.0. Tukey’s post hoc test was used in this study to analyze the data. Experiments were conducted in triplicate. * *p* < 0.05 vs. ACN treatment.

## 3. Results

### 3.1. Constitutive STAT3 Phosphorylation Inhibited by ACN within HepG2 Cells

This study examined the ability of ACN to modulate constitutive STAT3 activation in HepG2 cells. Several types of cancer cells, including hepatocellular carcinoma, have been linked to constitutive STAT3 activation, for example, via proliferation, survival, invasion, and angiogenesis [26,27]. The structure of ACN is given in Figure 1A. ACN curtailed STAT3 activation in dose-dependent manner (Figure 1B). Maximal inhibition was found to take place at 15 µmol/mL. In addition, STAT3 protein levels were found to have been affected by ACN. The inhibition of STAT3 activation according to incubation time with ACN was then characterized can (Figure 1C). ACN-induced inhibition was also found to time-dependent, with maximal inhibition at about 4 h. STAT3 protein expression was not shown to be impacted. ACN was observed to be stronger in the context of HepG2 cells, whose supplementation with different ACN dosages reduced the proliferation of cells, and their elongation was also curtailed, indicating the anticancer potential of ACN (Figure 1D).

### 3.2. STAT3 Binding Inhibition

DNA binding, the subsequent modulation of gene transcription, and STAT3 phosphorylation can be dependent on upstream tyrosine kinases, such as c-Src kinase and members of the JAK family. Phosphorylated STAT3 binds strongly to DNA and enhances the transcription of genes crucial to cell growth, which leads to the development and growth of tumors [28]. Next, we needed to ascertain the ability of ACN to modulate the DNA-binding activity of STAT3, for which we analyzed nuclear extracts made from HepG2 cells using an assay kit. As per the findings, ACN curtailed the aforementioned activities of DNA binding in a time-dependent manner, suggesting that ACN is indeed capable of abrogating DNA-binding ability of STAT3 (Figure 1E).

### 3.3. ACN Induces DNA Fragmentation

The action of ACN on significant cancer indicators, such as proliferation and resistance to apoptosis, was investigated using different HepG2 cancer cell lines. ACN also caused a substantial increase in DNA fragmentation in hepatic cells (Figure 1F). These findings indicate that ACN possesses an anticancer effect.

### 3.4. ACN Inhibits Cell Invasion

The influence of ACN on chemoinvasion was examined in HepG2 cells. ACN was found to considerably curtail the invasion of cells at doses of 10/15 μmol (Figure 1G), indicating the anti-invasive and antimigratory potentiality of ACN.

### 3.5. ACN Inhibits Il-6 Inducible STAT3, JAK2, and Akt Phosphorylation

IL-6 is an activator of Stat3 and is elevated in a variety of cancers [29,30,31]. To determine whether ACN could curtail STAT3 phosphorylation induced by IL-6, we treated Huh-7 cells with low concentrations of IL-6 across varying time frames and investigated whether STAT3 phosphorylation was induced by IL-6 treatment for 5 min. The maximal phosphorylation was found to occur between 10 and 30 min (Figure 2A). The first activation was seen at a dose of 15 ng/mL (Figure 2B). Moreover, ACN was found to suppress Akt/JAK2/STAT3 phosphorylation induced by IL-6 (Figure 2C). Cell exposure to ACN exceeding 4 h was found to be sufficient for completely inhibiting phosphorylation.

### 3.6. ACN Suppresses c-Src, JAK1, and JAK2 Activation

In normal cells, STAT3 activation is transient, but it plays a huge role in organ development and cell proliferation. However, in many types of tumors, STAT3 becomes constitutively active and is also mediated through the activation of JAK. c-Src kinase is also involved in STAT3 phosphorylation [32,33]. Since it is reportedly also possible to activate STAT3 via the constitutive activation of the soluble tyrosine kinases of the Src kinase family, we aimed to understand whether ACN impacts the constitutive activation of Src kinases across HepG2 cells. As per our findings, HepG2 did suppress constitutive phosphorylation as mentioned in the previous section (Figure 3A,B,C). When the conditions remained the same, there was no change in the levels pertaining to JAK1/JAK2 (nonphosphorylated). To confirm whether the inhibition of STAT3 phosphorylation induced by ACN is caused by phosphatase activation, HepG2 cells were treated with sodium pervanadate, an inhibitor of tyrosine phosphatase, which abolished the inhibition of STAT3 activation (Figure 3D), thus indicating the involvement of tyrosine phosphatases in the inhibition of STAT3 activation.

### 3.7. Depletion of STAT3 in the Nuclear Pool of HCC Cells

Since nuclear translocation plays a key role in the functionality of transcription elements and there is no clarity on whether its phosphorylation must be mandatorily used for STAT3’s nuclear transport along with the oncogenic functionalities [34], ACN was found to suppress the translocation of STAT3 to the nuclei of HepG2 cells (Figure 3E).

### 3.8. The Role of STAT3 in ACN-Induced Tumor Sphere Suppression

Further, to demonstrate the involvement of STAT3 in the ACN-induced inhibition of cell growth and proliferation, we genetically knocked down STAT3 and determined the expression of proteins associated cell growth regulation as well as proliferation; our analysis of the results supports the role of STAT3 in HepG2 cell growth (Figure 3F,G).

### 3.9. ACN Downregulates the Expression of Cell Survival Proteins

The activation of STAT3 supposedly controls the expression of numerous gene products; these products are found to play a role in the survival, angiogenesis, and proliferation of cells [35]. According to our findings, treatment with ACN modulates survivin, Mcl-1, Bcl-2, and Bcl-xL, which are anti-apoptotic proteins; cyclin D1, a regulator of cellular cycles (survivin); and VEGF (Figure 4A).

### 3.10. ACN Suppresses Cell Survival Associated Changes in Gene Expression

BCL-2 and VEGF were assessed using real-time PCR to determine the possible mechanism by which ACN influences the cell survival proteins and the expression of cyclin D1. The cells were maintained with ACN 15 for different time periods, and cyclin D1, BCL-2, and VEGF levels were measured at the RNA. We noticed a time-reliant reduction in cyclin D1, BCL-2, and VEGF and Snail protein expression in ACN-treated HepG2 cells (Figure 4B).

### 3.11. ACN Attenuates NF-κB Activation in HepG2 Cells

Next, we investigated the impact of ACN on NF-ĸB signaling cascade. The exposure to ACN for varying time intervals attenuated IκBα phosphorylation, an event that was a pre-requisite for p65 nuclear translocation and subsequent phosphorylation. ACN treatment also dampened constitutive p65 phosphorylation, as observed in a time kinetics study (Figure 4C). As evidenced in Figure 4C (top panel), ACN attenuates IKK activation, which confirms that ACN exerts its NF-κB inhibitory effects by interfering with IKK phosphorylation.

### 3.12. ACN Inhibits HCC Cell Proliferation and Apoptosis

We investigated whether ACN suppresses HCC cell proliferation via the MTT method. As per Figure 5A,B, ACN does curtail the cellular proliferation of Huh-7 as well as HepG2. We also examined whether the suppression of STAT3 (constitutively active) within HepG2 cells results in apoptosis. Pro-caspase-3 was found to be activated in a time-dependent manner in the above case (Figure 5C). Downstream caspase-3 activation, meanwhile, resulted in binding between a 116 and 85 kDa fragment, thereby indicating ACN’s ability to induce apoptosis reliant upon caspase-3.

### 3.13. ACN Induced Apoptosis in HepG2 Cells

Cell apoptosis was examined using TUNEL assay. Figure 5D illustrates that ACN (10 and 15 μmol) induces TUNEL-positive nuclei in HepG2 cells. Therefore, ACN treatment can potentially inhibit cell proliferation in HepG2 cells.

## 4. Discussion

The persistent activation of STAT3 denotes a major hallmark as far as malignant cancers are concerned. Thus, the therapeutic endeavor to target STAT3 is meritorious for treating cancers, and various strategies have been postulated in this regard. To begin with, preventing STAT3 from engaging with other key proteins is reported to promote STAT3 signaling. By focusing upstream of STAT3, it may be possible to develop inhibitors of kinase or inhibitors of cell surface receptors that regulate STAT3 [36,37]. In this context, Src/JAK inhibitors are presently undergoing clinical trials. Having said that, off-target toxicity causing inflammation to other STAT family members is known to limit such approaches. While a few novel inhibitors have emerged that act directly on STAT3, they are yet to undergo clinical trials. Against this backdrop, the current study is aimed at ascertaining whether ACN can partially demonstrate anticancer efficacy by abrogating the pathway of STAT3 signaling across HCC cells. As per previous studies, kinase activities of JAK2/Src work in tandem to mediate constitutive STAT3 activation [38]. As per our observations, ACN does show the ability to disrupt this coordinated involvement through tyrosine phosphorylation of STAT3. Furthermore, ACN has been shown to suppress IL-6-induced activation of all the aforementioned inhibitors in our study, which suggests that it also inhibits STAT reporter activity and nuclear translocation induced by EGF. The implication is that the flavone can potentially demonstrate its impact on inducible as well as constitutive STAT3activation via various means, in addition to curtailing the activation of inhibitors mentioned in previous sections. We also observed that ACN could affect the protein products of genes (regulated by STAT3), leading to a significant potential for apoptosis in HCC cells. After examining the hypothesis in a virtual system of predicting tumors, the findings suggest that ACN does inhibit STAT3 activation in human carcinoma cells. Such a mechanism has been experimentally proven. STAT3 inhibition by an Src inhibitor leads to the downregulation of Mcl-1 gene expression in melanoma cells. Moreover, STAT3 signaling activation induces survivin gene expression, which leads to resistance to apoptosis in cells associated with breast cancer in humans [39]. Bcl-2 is also capable of preventing cell death caused by numerous chemotherapeutic agents [40,41]. For this reason, downregulating Bcl-2′s expression probably highlights the ability of ACN to induce apoptosis in cancerous cells. Downregulated VEGF expression is consistent with a recent study that shows that HCC growth as well as the in vivo angiogenesis of tumors is inhibited by targeting VEGF [42].

STAT3 that is constitutively active is found in many types of cancer, such as neck/head [43], lymphomas [44], and multiple myeloma [45], as well as leukemia [46]. Therefore, its suppression across HCC cells could suggest that the inhibitor of STAT3 may also suppress STAT3 (constitutively activated) in cells affected by other kinds of cancer. In the past, ACN has been shown to be capable of inhibiting NF-κB activation [47]. However, there is no clarity on whether ACN’s suppression of STAT3 activation is associated with the curtailment of NF-κB activation. The p65 subunit of NF-κB reportedly engages with STAT3. However, the activation of NF-κB and STAT3 takes place in dissimilar cytokines; while TNF potently activates the former, IL-6 strongly activates STAT3. A previous study also demonstrated the potential ability of erythropoietin to activate NF-κB by activating JAK2 kinase [48]. Therefore, the inhibition of JAK2 kinase activation could be pivotal to suppressing STAT3 activation.

## 5. Conclusions

Overall, the results of our experiments suggest that in HCC cells, the pro-apoptotic/antiproliferative attributes of ACN can be attributed to its suppression of STAT3 activation, thereby making a strong case for the further exploration of the potential of ACN to enhance the efficacy of treatment aimed at HCC patients while also lowering the associated toxicity.

## Figures and Tables

**Figure 1 molecules-27-05361-f001:**
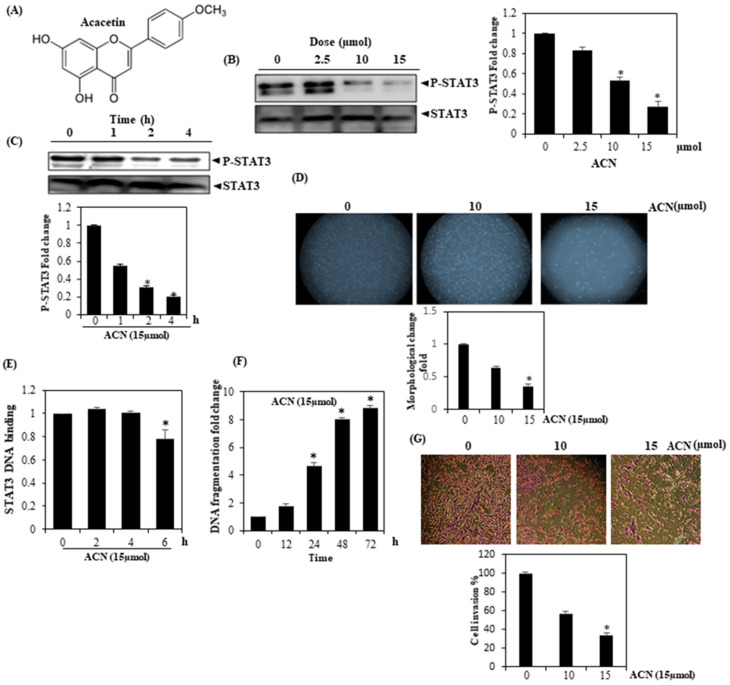
(**A**) Constitutively active STAT3 is inhibited by ACN. (**B**) p-STAT-3 levels are inhibited by ACN in a time-dependent manner. After treating HepG2 cells with indicated concentrations of ACN for 4 h, preparation of whole cell extracts was undertaken; protein weighing 50 mg was resolved over SDS-PAGE gel (10%) before being transferred to nitrocellulose membranes. (**C**) After treating HepG2 cells with ACN (15 µmol/mL) for the indicated amounts of time, Western blotting was performed. STAT3 antibody was used to strip and again probe the blots to confirm that there were no discrepancies in protein loading. (**D**) Morphological examination of cells incubated with 15 µmol/mL ACN for 24 h. (**E**) DNA-binding assay using nuclear extract protein (20 µg) and (**F**) DNA fragmentation of cells treated with 15 µmol/mL ACN for the indicated time points. (**G**) Observation of HepG2 cell invasion. The findings signify three experiments carried out independently (* *p* < 0.05).

**Figure 2 molecules-27-05361-f002:**
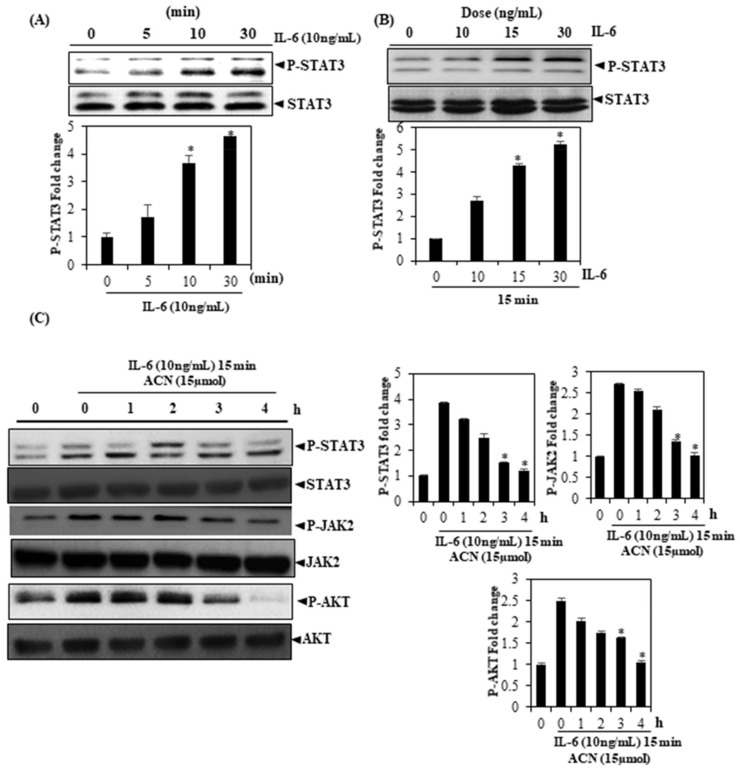
(**A**) ACN is shown to downregulate p-STAT3 (induced by IL-6) levels across HCC cells. (**B**) After treating Huh-7 cells with IL-6 for the specified time, the preparation of entire cell extracts was undertaken. Western blotting was used for detecting p-STAT3, and the STAT3 antibody was used again. A probe of these blots, following stripping, confirms that protein loading was equal. After Huh-7 cells were exposed to ACN (15 µmol/mL) for the specified amounts of time, they were stimulated by IL-6 for 15 min. (**C**) After preparing the aforementioned cell extracts, we analyzed them using Western blotting and then stripped/re-investigated the blots using STAT3 antibody. AKT and JAK2 antibodies were used to follow the same procedure to ascertain equality in protein loading. The findings signify three experiments carried out independently (* *p* < 0.05).

**Figure 3 molecules-27-05361-f003:**
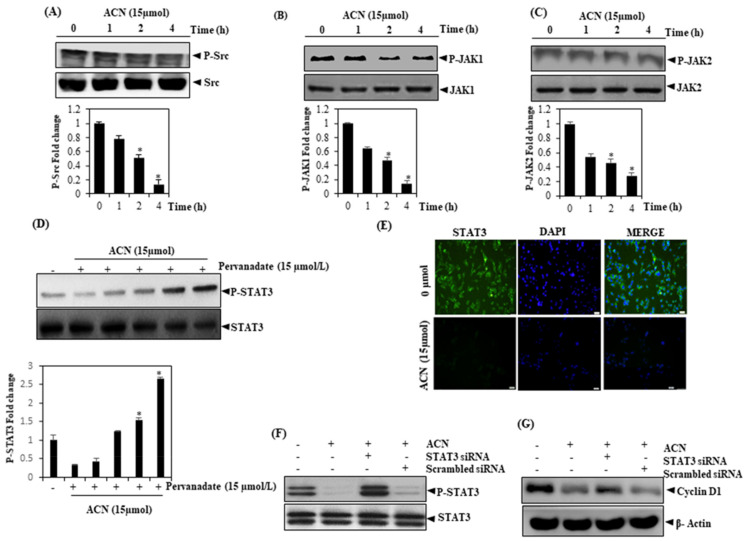
(**A**) Inhibition of constitutively active STAT3 by ACN. After exposing HepG2 cells to 15 µmol/mL of ACN, extracts of the entire cells were taken. Then, 30 mg aliquots of the extracts were resolved on SDS-PAGE gel (10%) before being transferred to nitrocellulose membranes, eventually being examined using the p-Src antibody, and blots were in turn stripped and re-investigated to confirm equal protein loading. (**B**) ACN suppresses p-JAK1 levels in a time-dependent manner, and p-JAK1 is examined using an pJAK1 antibody. JAK1 antibody was stripped and re-investigated for ascertaining equality in protein loading. (**C**) p-JAK2 levels were also investigated using a p-JAK2 antibody and found to be suppressed in a time-dependent manner. This step assumes significance. (**D**) The suppression of p-STAT3 by ACN is reversed by pervanadate. Extracts of the cells were taken after HepG2 cells were treated with ACN (15 µmol/mL) as well as pervanadate in the concentrations indicated before for 4 h of which 30 mg was resolved over SDS-PAGE gel (7.5%), before transfer to nitrocellulose membranes for the characterization of STAT3/p-STAT3. (**E**) ACN curtails STAT3 translocation to the cell nucleus. After HepG2 cells were incubated in the presence or absence of ACN (15 µmol/mL) for 4 h, the intracellular distribution of STAT3 was assessed. HepG2 cells were transfected transiently with siRNA-STAT3 and treated with 15 μmol of ACN. Protein was isolated and analyzed for the measurement of p-STAT3 and cyclin D1 protein by Western blot analysis (**F**,**G**). The findings signify three experiments carried out independently (* *p* < 0.05).

**Figure 4 molecules-27-05361-f004:**
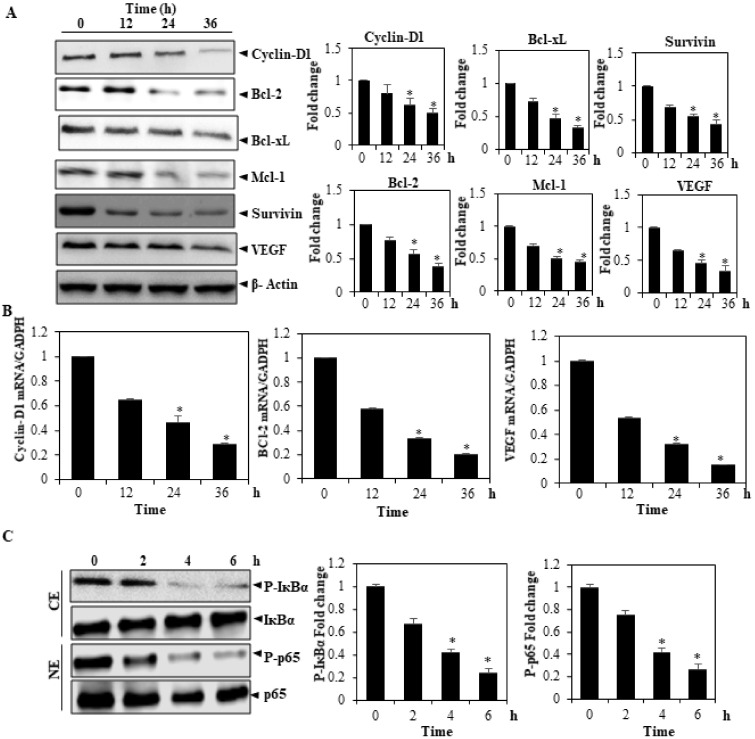
(**A**) can inhibit the production of proteins from genes regulated by STAT3 involved in survival, angiogenesis, and proliferation. After treating HepG2 cells with 15 µmol/mL of ACN for the indicated amounts of time, extracts of the entire cells were taken, of which 30 mg was resolved over SDS-PAGE gel (10–15%) and transferred to membranes, which were sliced in accordance with molecular weight. They were then investigated using the following antibodies: Mcl-1VEGF, Bcl2, Mcl 1, D1, and Bcl-xl. The β-actin antibody was used to strip/re-investigate these blots for ensuring equal protein loading. (**B**) After treating HepG2 cells with ACN, the cells were harvested following treatment, followed by the extraction of RNA samples. Notably, 1 mg of these RNA extracts was used to conduct reverse transcription for generating the corresponding cDNA. PCR was performed in real time for measuring relative mRNA quantities (**C**). All gene products were characterized using the aforementioned antibodies, with GAPDH being established as an endogenous control for normalization of samples. Sequence Detection Software version 1.3 was used to analyze the results. Following normalization using GAPDH and verifying the threshold cycle difference between untreated and treated cells, we obtained the relative gene expression levels. Cytosolic nuclear fractionation and Western blot analysis of the effects of ACN on Pp65 and p-IkBα levels are also illustrated. The findings signify three experiments carried out independently (* *p* < 0.05).

**Figure 5 molecules-27-05361-f005:**
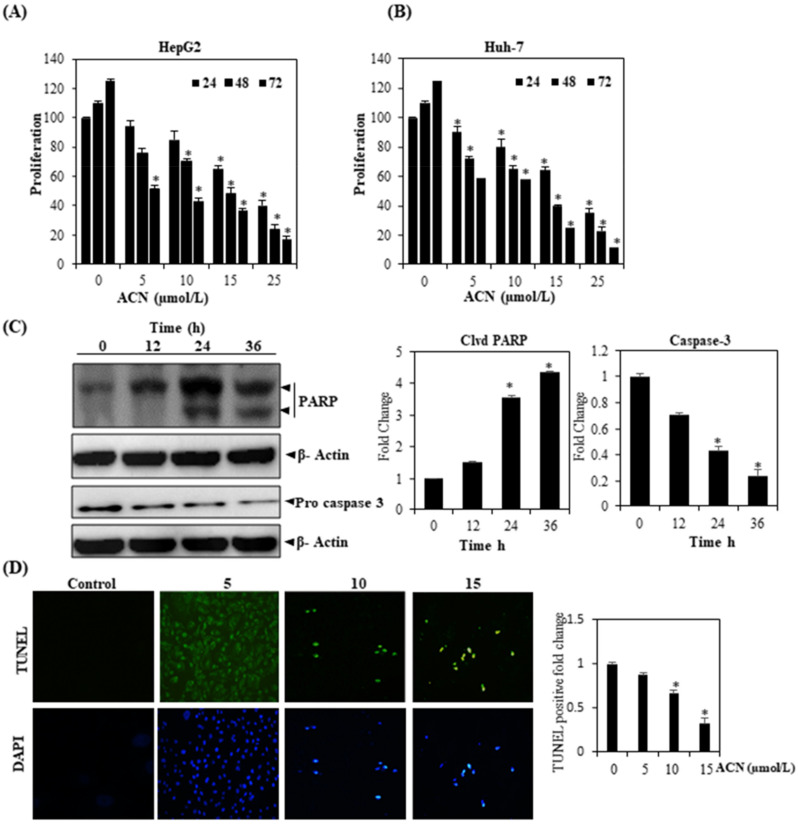
ACN suppresses proliferation, activating caspase-3. After Huh-7 and HepG2 were introduced in triplicate, they were exposed to ACN in predetermined concentrations. An MTT assay was then performed after 1, 2, and 3 days for examining cellular proliferation in (**A**) HepG2 and (**B**) Huh-7 cells. (**C**) After HepG2 cells were treated with ACN (15 µmol/mL) for predetermined amounts of time, cell extracts were prepared and extracted on SDS-PAGE gel prior to undergoing Western blotting using antibodies against procaspase-3 and PARP. β-Actin antibody was used and re-investigate these blots following stripping to confirm equal protein loading. (**D**) DNA fragmentation was assessed using transferase-mediated dUTP-fluorescein nick end labeling (TUNEL). A microscope was used to observe three fields of cells containing average numbers related to apoptotic positive cells. The findings signify three experiments carried out independently (* *p* < 0.05).

## Data Availability

The data that support the findings of this study are available from the corresponding author upon reasonable request.

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
