# Peer review of "Acacetin Inhibits Cell Proliferation and Induces Apoptosis in Human Hepatocellular Carcinoma Cell Lines"

_molecules, 2022, doi:10.3390/molecules27175361_

Round 1

Reviewer 1 Report

Dear Madam/Sir,

The authors failed to respond satisfactorily to the Reviewers’ comments on the previous manuscript. Even worse, the name of the studied compound was misspelled in the title, suggesting this manuscript was carelessly prepared.

Comments on this revised manuscript are listed as follows.

1. The most serious flaw of this revised manuscript is located in the title. The authors need to clarify the true identity of the compound they studied. In other words, whether they studied the anti-HCC effect of “Cacetin” or “Acacetin”.

2. The authors failed to respond satisfactorily to the Reviewers’ comments on the previous manuscript. These include:

(1) the authors did not indicate what is the residue of STAT3 to be phosphorylated for STAT3 activation.

(2) The authors did not demonstrate the essential role of STAT3 blockage in the inhibitory effects of acacetin on the proliferation, survival, motility and invasiveness in both HepG2 and Huh cells. In other words, the authors have to test the effect of acacetin on the HCC cells with ectopic expression of dominant-active STAT3.

(3) The question raised in the Reviewers’ comments about the logical errors found in Figure 3F and 3G remain unanswered.

(4) The authors have to reveal the methodology to evaluate the DNA-bonding activity of STAT3.

Author Response

please see the attach file

Reviewer 2 Report

The manuscript in the present form has improved, but there are still several problems in terms of clarity and English language in the presentation of data. Before acceptance, an extensive editing of English language throughout the main text and in figure legends is necessary.

Author Response

Reviewer 2

The manuscript in the present form has improved, but there are still several problems in terms of clarity and English language in the presentation of data. Before acceptance, an extensive editing of English language throughout the main text and in figure legends is necessary.

According to the reviewer’s suggestion, full manuscript has been edited by MDPI English editing.

Reviewer 3 Report

This manuscript used a functional proteomics tumor pathway technology platform and multiple HCC cell lines to investigate the effects of acacetin (ACN) on STAT3 activation, protein kinases, phosphatases, and products of STAT3-regulated genes, and apoptosis. Authors found that ACN can inhibit STAT3 activation in a dose- and time-dependent manner in HCC cells. Up-stream kinases c-Src, JAK 1, and JAK 2 were also inhibited. ACN was found to suppress the protein expression of genes involved in proliferation, survival, and angiogenesis via STAT3 inhibition.

The manuscript matches the criteria for publication in Molecules. I would suggest to published here. However, the following minor revisions are necessary;

1. Since
ACN appears to be a novel STAT3 inhibitor and involved in proliferation, survival, and angiogenesis via STAT3 inhibition in HCC. Did the authors check ACN direct binding assay with STAT3? I would suggest authors include any binding assay to confirm ACN binds with STAT3, like SPR or pulldown assay.

2. One typo of writing that needs to be corrected, “Cacetin” in the title should be changed to “Acacetin”.

Author Response

Reviewer 3

This manuscript used a functional proteomics tumor pathway technology platform and multiple HCC cell lines to investigate the effects of acacetin (ACN) on STAT3 activation, protein kinases, phosphatases, and products of STAT3-regulated genes, and apoptosis. Authors found that ACN can inhibit STAT3 activation in a dose- and time-dependent manner in HCC cells. Up-stream kinases c-Src, JAK 1, and JAK 2 were also inhibited. ACN was found to suppress the protein expression of genes involved in proliferation, survival, and angiogenesis via STAT3 inhibition.

The manuscript matches the criteria for publication in Molecules. I would suggest to published here. However, the following minor revisions are necessary;

Thank you for your valuable comments.

  1. Since ACN appears to be a novel STAT3 inhibitor and involved in proliferation, survival, and angiogenesis via STAT3 inhibition in HCC. Did the authors check ACN direct binding assay with STAT3? I would suggest authors include any binding assay to confirm ACN binds with STAT3, like SPR or pulldown assay.

Yes, we did.  Our university we don’t have Surface Plasmon Resonance instrument. We have ordered the Dynabeads™ Co-Immunoprecipitation Kit from Thermofisher Scientific (Cat no: 14321D) unfortunately Saudi customs to block the kit, they keep normal room temperature more than 20 days, we got approval from ministry, released the kit. Our bad time the kit doesn’t work.

Not only this kit and also nude mice, they are not allowed.

We are really sorry. In future first we get approval from ministry and then order the reagents. 

  1. One typo of writing that needs to be corrected, “Cacetin” in the title should be changed to “Acacetin”.

Sorry for that mistake, our revised manuscript we have corrected title of the manuscript.

We thank the reviewers for their valuable suggestions to highlight the significance of this review and improve the review article to meet the journal’s standards. We hope that this revised manuscript is now better suited for publication in your esteemed journal.

Round 2

Reviewer 1 Report

1. The authors still did not address my previous comment (3) regarding the role of the STAT3 inhibition in acacetin-induced inhibition of malignant phenotype in both HCC cell lines.

2. The authors still did not fix the logical errors present in Figure 3F. Why are the p-STAT3 and total STAT3 still present in STAT3 siRNA-treated cells?

This manuscript is a resubmission of an earlier submission. The following is a list of the peer review reports and author responses from that submission.

Round 1

Reviewer 1 Report

The manuscript by Peramaiyan Rajendran and colleagues is a very messy summary of the anticancer effects induced by the natural compound Acacetin in HCC cells. A major crucial issue is about English language and style, as too many inconsistencies were found, thus the entire manuscript, although potentially interesting, is quite unreadable in the present form. Specifically, there are too typos throughout the text, many details in describing data and the corresponding figures are lacking, whereas there are too many technical details in figure captions. Indeed, text should be substantially reorganized, as in the case of “Discussion” section.

Moreover, the experiments are alternately conducted in HCC or Huh-7 cells, thus some doubts on the reliability of the data still remain.

Reviewer 2 Report

This manuscript described the inhibitory effect of acacetin, a flavonoid compound, on the proliferation, viability, and invasiveness of human hepatocellular carcinoma (HCC) cell lines HepG2 and Huh7 cells. Furthermore, the mechanisms of acacetin-mediated antiproliferation, apoptosis, and invasiveness were associated with the suppression of STAT3 activation. Comments for this manuscript were listed below:

Major comments:

  1. All of the acacetin-induced impacts, including suppression of STAT3 activation, apoptosis induction, and inhibition of migration/invasion, must be demonstrated in both HepG2 and Huh 7 cell lines.
  2. The authors must clearly justify how to define STAT3 activation in this study in the Introduction and Results
  3. The authors must reveal the residue of phosphorylation of STAT3 shown in Figures 1~3. Likewise, the phosphorylated residues of JAK1, JAK2, and SRC addressed in Figure 3 need to be defined.
  4. Results shown in Figure 3F and 3G are confusing. Why STAT3 siRNA plus acacetin co-treatment led to an increase in the levels of phosphorylated STAT3?
  5. In Figure 3, the logic to address the role of STAT3 in the acacetin-induced anti-HCC effect in this study is perplexing. The logical approach is to examine whether blockage of acacetin-induced STAT3 suppression may counteract the inhibitory effect of acacetin-mediated antiproliferation, apoptosis, and invasiveness.
  6. At Line 73, the authors need to justify how the TransAM NF-kappaB assay kit can measure the STAT3’s DNA-binding activity.
  7. Given no in vivo study was performed, the title of this study must be rephrased as …” in hepatocellular carcinoma cell lines”.

Minor comments:

  1. The comma shown in the title must be deleted.
  2. The statements shown at Lines 19 and 22~24 are confusing and therefore must be rephrased.
  3. Please show the quantification results of Figures 1G and 1H.
  4. Figure 4C, please define the Y-axes of both bar charts.
  5. Figure 5D, please include a scale bar.